**Data Availability Statement:** Data Availability Statement: The Illinois, Indiana, and Wisconsin mortality data on which we rely was obtained from

# Evidence on COVID-19 Mortality and Disparities Using a Novel Measure, COVID excess mortality percentage: Evidence from Indiana, Wisconsin, and Illinois

Vladimir Atanasov[1], Natalia Barreto[2], Lorenzo Franchi[3], Jeff Whittle[4], John Meurer[4], Benjamin W. Weston[4], Qian (Eric) Luo[5], Andy Ye Yuan[6]*, Ruohao Zhang[7], Bernard Black[8]

1 William & Mary, Mason School of Business, Williamsburg, Virginia, United States of America, 2 University of Illinois, Champaign-Urbana, Illinois, United States of America, 3 Northwestern University, Evanston, Illinois, United States of America, 4 Medical College of Wisconsin, Milwaukee, Wisconsin, United States of America, 5 George Washington University, Washington, DC, United States of America, 6 Northwestern University, Pritzker School of Law, Evanston, Illinois, United States of America, 7 Pennsylvania State University, State College, Pennsylvania, United States of America, 8 Northwestern University, Pritzker School of Law and Kellogg School of Management, Evanston, Illinois, United States of America

* andyyuan@law.northwestern.edu

## Abstract

COVID-19 mortality rates increase rapidly with age, are higher among men than women, and vary across racial/ethnic groups, but this is also true for other natural causes of death. Prior research on COVID-19 mortality rates and racial/ethnic disparities in those rates has not considered to what extent disparities reflect COVID-19-specific factors, versus preexisting health differences. This study examines both questions. We study the COVID-19-related increase in mortality risk and racial/ethnic disparities in COVID-19 mortality, and how both vary with age, gender, and time period. We use a novel measure validated in prior work, the COVID Excess Mortality Percentage (CEMP), defined as the COVID-19 mortality rate (Covid-MR), divided by the non-COVID natural mortality rate during the same time period (non-Covid NMR), converted to a percentage. The CEMP denominator uses Non-COVID NMR to adjust COVID-19 mortality risk for underlying population health. The CEMP measure generates insights which differ from those using two common measures–the COVID-MR and the all-cause excess mortality rate. By studying both CEMP and COVID-MRMR, we can separate the effects of background health from Covid-specific factors affecting COVID-19 mortality. We study how CEMP and COVID-MR vary by age, gender, race/ethnicity, and time period, using data on all adult decedents from natural causes in Indiana and Wisconsin over April 2020-June 2022 and Illinois over April 2020-December 2021. CEMP levels for racial and ethnic minority groups can be very high relative to White levels, especially for Hispanics in 2020 and the first-half of 2021. For example, during 2020, CEMP for Hispanics aged 18–59 was 68.9% versus 7.2% for non-Hispanic Whites; a ratio of 9.57:1. CEMP disparities are substantial but less extreme for other demographic groups. Disparities were generally lower after age 60 and

the respective state Departments of Health and Vital Statistics, and is covered by data use agreements which prevent us from publicly releasing it. If anyone is interested in acquiring this dataset, please contact State of Wisconsin Department of Health Services Division of Public Health Data Resource Center, dhsdphdataresourcecenter@dhs.wisconsin.gov.

**Funding:** Research reported in this publication was supported by the National Center for Advancing Translational Sciences of the National Institutes of Health under Award Number UL1TR001436 to the Medical College of Wisconsin Clinical and Translational Science Institute of Southeast Wisconsin. The content is solely the responsibility of the authors and does not necessarily represent the official views of the NIH. We received no other external funding. Internal funding for the authors was limited to salary and, for Professor Black, a research budget that is not specific to this project. He received research support from the Ray Garrett Jr. Memorial Fund, the David S. Ruder Corporate Research Fund, the Arthur R. Seder, Jr., Corporate Research Fund, and the Northwestern Pritzker School of Law Faculty Research Program. The funders had no role in study design, data collection and analysis, decision to publish, or preparation of the manuscript. The draft was reviewed by the Wisconsin Department of Health, in accordance with our Data Use Agreement for the Wisconsin data that we rely on. to ensure that it correctly identified their role as a data provider and did not disclose personally identifiable information.

**Competing interests:** The authors have declared that no competing interests exist.

declined over our sample period. Differences in socio-economic status and education explain only a small part of these disparities.

## Introduction

The COVID-19 mortality rate (COVID-MR, the fraction of the population who died from COVID-19) is small for the young, but rises rapidly with age and is higher for men than for women and for racial and ethnic minorities than for non-Hispanic White Americans [1–4]. However, mortality rates from other natural causes are also associated with age, gender, and race/ethnicity [5]. Similarly, lower socio-economic status (SES) and less education correlate with both higher COVID-19 mortality [6], and higher mortality from other natural causes [7]. Prior work on COVID-19 mortality rates and the factors associated with higher mortality rates typically studies either COVID-MR or all-cause excess mortality ("excess mortality," defined as total mortality from all causes, relative to predicted mortality based on pre-COVID rates) [1–3,6–13]. This prior work generally does not separate differences in COVID-19 mortality rates that reflect differences in underlying health status and thus mortality risk, from COVID-specific differences [8–10]. Moreover, the impact of COVID-19 on different demographic groups has changed substantially over the pandemic period [11].

While prior work shows that non-White populations have higher COVID-19 mortality rates, there is limited understanding of the reasons for these disparities. Some likely reflect pre-existing health differences [14]. However, some groups, notably Hispanics and Asians, have lower background natural mortality rates than Whites, and thus would be expected to have lower COVID-19 mortality risk if infected, assuming similar treatment. The effect of COVID-specific risk on mortality could be masked by measures which do not account for background risk. COVID-MR and excess mortality measures also draw the mortality numerator from death records, but the population denominator from survey data (Census or American Community Survey); the two sources may define minority groups differently, leading to biased estimates.

We seek to separate the influence on COVID-19 mortality of background health versus pandemic-specific factors such as exposure to infection, variation in adopting risk-prevention measures, and any COVID-specific differences in health care once infected. To do so, we study a novel measure, the COVID Excess Mortality Percentage (CEMP). CEMP is defined as COVID-19 deaths divided by Non-COVID natural deaths, expressed as a percentage. We refer to CEMP as measuring "excess" mortality because it measures the extra mortality due to COVID-19, assuming that the COVID pandemic did not affect rates for other natural causes of death. The CEMP denominator uses non-Covid natural mortality rates for a demographic group to proxy for underlying health, and thus for COVID-19 mortality risk if infected. We measure CEMP for population groups defined by age, gender, and race/ethnicity. We previously validated this measure as providing a good proxy for background health by showing that pre-pandemic natural mortality rates is strongly correlated with COVID-19 mortality rates in the pre-vaccine period [15–18]. CEMP provides a measure complementary to COVID-MR and excess mortality, which lets one assess to what extent the differential impact of COVID-19 on the elderly, men, the poor, and racial/ethnic minorities reflects differences in background health versus Covid-specific factors. We also assess the correlation between CEMP and education, and the correlation of CEMP with socio-economic status measured at the zip-code level (zip-SES). We study both the pre-vaccine period and periods when vaccines were available,

and compare racial/ethnic disparities using CEMP versus COVID-MR as the outcome of interest.

We find very large disparities in COVID-specific risk across racial/ethnic groups, especially early in the pandemic, particularly for Hispanics. The observed disparities vary substantially with age and across time periods during the pandemic. For example, during 2020, CEMP for Hispanics aged 18–59 was 68.9% versus 7.2% for non-Hispanic Whites; a ratio of 9.57:1. CEMP disparities are substantial but less extreme for other minorities. Disparities were generally lower after age 60, and declined over our sample period. Differences in socio-economic status and education explain only a modest part of these disparities. Asian/White disparities are also substantial during the pre-vaccine period, but were obscured by the COVID-MR measure, which does not account for Asians' lower background mortality risk.

## Data and methods

### Data

We obtained de-identified mortality records for all adult decedents in Wisconsin and Indiana for 2020 through June 30, 2022 (279,000 natural cause deaths, including 34,000 COVID-19 deaths, from adult population of 9.7 million), and for Illinois through December 31, 2021 (213,000 natural cause deaths, including 28,000 COVID-19 deaths, from adult population of 9.9 million). We studied these states because we have access to individual-level death records, including 5-digit residence zip code, education, age, marital status, manner of death, and text fields indicating primary cause of death, contributing causes, and other significant conditions. We studied all deaths due to natural causes, defined as all deaths except those with manner-of-death of accident, homicide, or suicide. We used text analysis of the cause-of-death fields to identify natural deaths due to COVID-19 versus those with other causes. This approach attributed more deaths to COVID-19 than did the ICD-10 cause-of-death codes assigned by the National Center for Health Statistics (NCHS). See Appendix in S1 File for details. We did not have data on individual SES, but measured decedents' zip-SES, using quartiles of the Graham Social Deprivation Index [12]; we use zip-SES in multivariate logistic regressions. We obtained population data and zip-SES in 2020 from the American Community Survey. The project was approved by the Medical College of Wisconsin Human Research Review Board. All the data were fully anonymized before we accessed them.

### Analysis

We define CEMP within cells defined by age group, gender, and race/ethnicity, within a specific time period, as:

$$CEMP = \frac{COVID-19\ deaths}{Natural\ deaths - COVID\ deaths} \tag{1}$$

The CEMP denominator (non-COVID natural deaths) serves as a proxy for underlying health.

We can validate the CEMP measure by examining the association between natural mortality rates and COVID-19 mortality rates during 2020, when vaccines were not yet available (Fig 1, Panel A). The Pearson correlation coefficient between Non-Covid-NMR in April-December 2019 and COVID-19 MR during early and late stages of the pandemic, within cells defined by age group, gender, race/ethnicity and state, ranges is 0.90 (Fig 1, Panel A) [15]. The high correlation is not surprising, because many specific risk factors for COVID-19 mortality (e.g., age, gender, obesity, diabetes) are also correlated with non-COVID mortality [13].

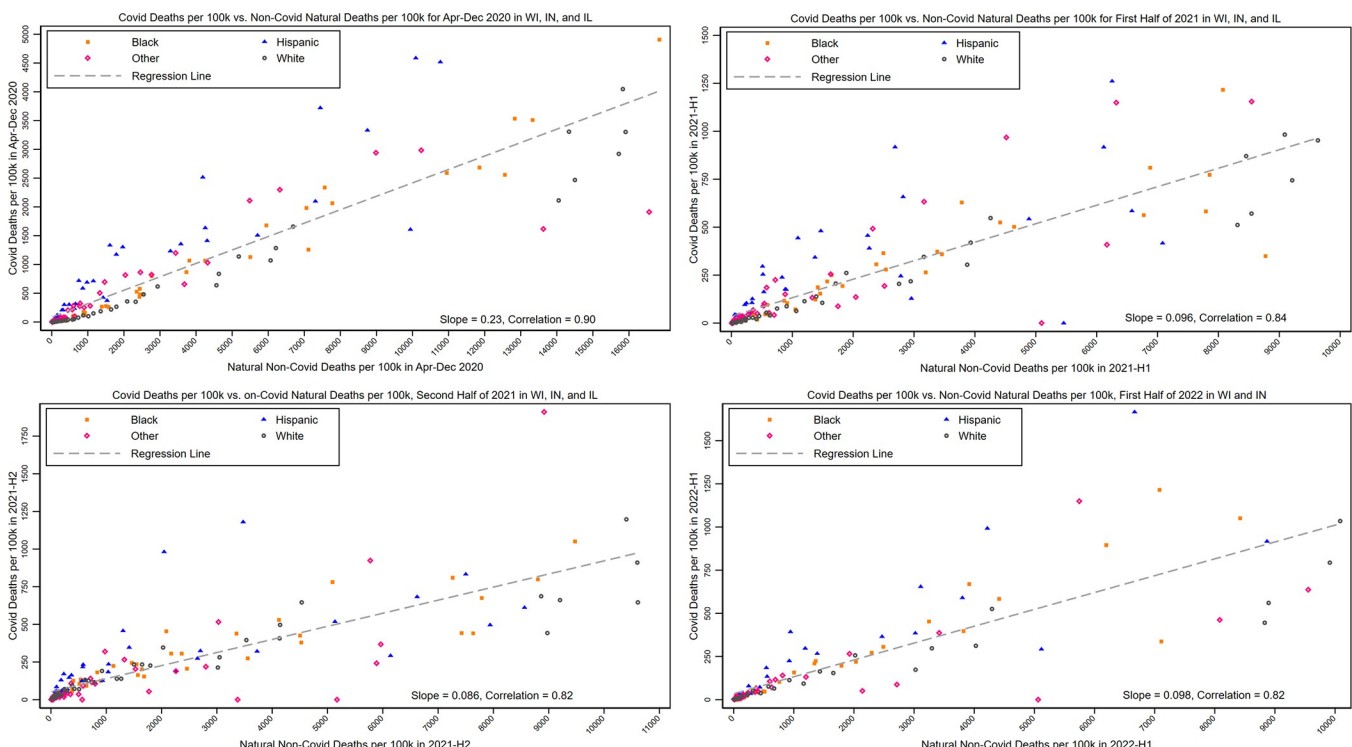

**Fig 1. Correlation between COVID-19 and Non-COVID natural mortality rates. Panel A.** Scatterplot of COVID-19 mortality rates versus Non-COVID natural mortality rates in Illinois, Indiana, and Wisconsin, expressed as deaths per 100,000 persons, over April-December 2020, for groups defined by state*age (18–39, 40–49, 50–59, 60–64, 65–69, 70–74, 75–79, 80–84, 85–89, 90–94, 95+)*gender*race/ethnicity, best-fit regression line, and Pearson correlation coefficient. Plot includes 168 bins (3 states*7 age groups*2 genders*4 race/ethnicities), best-fit regression line, and Pearson correlation coefficient. **Panel B.** Same for Early-Vaccine Period (1H-2021). **Panel C.** Same for Delta Period (2H-2021). **Panel D.** Plot for 1H-2022. Plot includes only Indiana and Wisconsin and has 112 bins (2 states *7 age groups*2 genders*4 race/ethnicities). **Panel A. Pre-Vaccine Period (April-December 2020). Panel B. Early Vaccine Period (January-June 2021). Panel C. Delta Period (July-December 2021). Panel D. Omicron Period (January-June 2022).**

We confirm the strong association in the Appendix in S1 File, in several ways. First, a similar association is seen for mortality *counts* within groups defined by age, gender, race/ethnicity, and zip-SES (Figure App-1 in S1 File), consistent with the association for mortality *rates* shown in Fig 1. Second, we confirm that there is strong association between COVID mortality rates and non-COVID mortality rates within each racial/ethnic group: 0.98 for Whites; 0.99 for Blacks; 0.92 for Hispanics; and 0.85 for other (Table App-11 in S1 File). Third, we measure the association in all three states between natural mortality counts *in 2019* (pre-pandemic, no natural mortality was not affected by the pandemic) and COVID-19 mortality counts in 2020 (Figure App-2 in S1 File). The correlations are 0.97 for Indiana; 0.98 for Wisconsin, and 0.93 for Illinois.

CEMP ignores any effect of COVID-19 infection on non-COVID mortality. To the extent that prior COVID-19 infection causes higher non-COVID mortality [19–22], CEMP will understate the full COVID mortality burden. However, once we code COVID-19 deaths based on death certificate text fields, we find no evidence of substantial undercounting in Indiana or Wisconsin. Appendix Figure App-3 in S1 File reports monthly natural non-COVID-19 and all natural deaths for Wisconsin and Indiana for 2017–June 2022, and for Illinois for 2017-December 2021. For the pandemic period, we also show predicted natural non-COVID deaths, based on linear extrapolation from 2017–2019 to the same calendar month during the pandemic period. For Wisconsin and Indiana, natural deaths (including COVID-19 deaths)

show two COVID-related peaks in late 2020 and late-2021-early 2022. Natural non-COVID-19 deaths do not have corresponding spikes, which would be expected if COVID deaths were significantly undercounted. Predicted natural non-COVID natural deaths (dashed line) are close to measured non-COVID natural deaths, sometimes higher or lower, but with no obvious pattern. Measured non-COVID natural deaths are never outside the 95% confidence interval (CI) (See Appendix Figure App-4 for CIs in S1 File).

For Illinois, natural-non-COVID deaths rise during peak COVID-mortality periods (Figure App-4, Panel C in S1 File). This is evidence consistent with undercounting of COVID-19 deaths, although it could also reflect, in part, medical care disruption during peak COVID periods. The rise in natural non-COVID deaths during peak COVID periods implies that CEMP will be underestimated. However, any underestimate will affect minority/White CEMP ratios only if the undercount differs across different racial/ethnic groups.

We report CEMP levels for adults aged 18+ within cells defined by age, gender, and race/ethnicity, for ages 18–39, 40–49, 50–59, 60–69, 70–79, 80–89, and 90+. We divide the decedents into non-Hispanic White ("White"), Black, non-Black Hispanic ("Hispanic") and Other (including Asian, Native American, and mixed race); our sample is too small to permit disaggregation of the "Other" group. We report CEMP ratios, within cells defined by age group and gender, for Black/White (defined as Black CEMP/White CEMP), and similar ratios for Hispanic/White, and Other/White. We do not study children due to their low COVID mortality. CEMP will be undefined for cells with no non-COVID natural deaths; we did not encounter this issue for our sample.

For decedents from natural causes, CEMP represents the odds of dying from COVID-19 versus other natural causes. The ratio of CEMPs for two different groups, say Hispanics and Whites, is an odds ratio, which can be recovered from logistic regression. We accordingly use logistic regression to compute p-values (reported in the text) and 95% confidence intervals (CIs) (reported in the Appendix in S1 File) for the CEMP ratios in Table 2.

We also conduct multivariate logistic regression analysis and report odds ratios for how race/ethnicity and other factors affect CEMP. We run regressions separately for persons aged 18–59 and those aged 60+ (reflecting evidence presented below that racial/ethnic disparities are higher for ages 18–59). The regression predictors are age group, race/ethnicity, gender, zip-SES, education, and marital status.

CEMP, as an outcome measure, controls for pre-existing health and thus pre-existing disparities in mortality rates. To separate the role of COVID-19-specific factors from background health in predicting overall COVID mortality, we also report COVID MR, non-COVID natural mortality rates (Non-COVID-NMR), and MR ratios (e.g., Hispanic COVID-MR/White COVID-MR). Within a given time period:

$$\text{CEMP} = \frac{COVID\ deaths}{Non-Covid\ natural\ deaths}$$

$$\text{COVID MR} = \frac{COVID\ deaths}{Population}$$

$$\text{Non-COVID NMR} = \frac{Non-Covid\ natural\ deaths}{Population}$$

CEMP, Non-Covid NMR, and COVID-MR are related by:

$$\text{CEMP} = \frac{COVID\ MR}{Non-COVID-NMR}$$

Comparing minority/White ratios measured using CEMP as the outcome to ratios measured using COVID-MR as the outcome allows assessment of the relative contribution of COVID-19 specific factors and background health to COVID-19 mortality.

We report results for four time periods: a "pre-vaccine" period from April-December 2020, an "early-vaccine" period from January to June, 2021 (1H-2021), when vaccines were becoming available, first to the elderly and healthcare workers, with first doses becoming generally available by April); a "Delta" period from July to December 2021 (2H-2021), when vaccines were widely available and Delta was the dominant COVID variant; and an "Omicron" period from January to June 2022, when Omicron was the dominant virus variant. We begin analysis in April 2020. COVID-19 was declared a national emergency in mid-March 2020, but COVID-19 mortality relative to other natural causes was low for March 2020 as a whole. Also, mortality counts for March 2020 are unreliable due to limited access to Covid-19 tests.

## Results

### Study population

Table 1 provides summary statistics for our sample of decedents. Men are a higher proportion of COVID decedents than of all decedents. Hispanics have non-COVID natural deaths well below their share of population, reflecting younger average age, yet COVID-19 deaths much closer to their share of population. A ratio for each racial/ethnic group of (% of COVID-19 deaths)/(% of Non-COVID natural deaths) provides a simple overall measure of pandemic-specific risk, relative to background mortality risk, across our full sample period:

Hispanic = (9.1%/3.24%) = 2.81
Black = (12.7%/11%) = 1.16
Other = (2.6%/1.9%) = 1.37
White = (75.6%/83.9%) = 0.90

Below, we will decompose these average levels by age group, time period, and (in the Appendix in S1 File) gender. Those decompositions are important. Still, these simple ratios begin to develop some regularities that will emerge from the more detailed analysis below. First, Hispanics faced much higher Covid-specific risk than other groups. Blacks and Other faced moderately elevated Covid-specific risk, but well below Hispanic risk. Second, high Black COVID-MR relative to Whites, known from prior work and confirmed here, reflects moderately elevated COVID-specific risk combined with substantially higher background risk.

### Pre-vaccine period

Table 2A to 2D presents data on COVID-MR, non-COVID NMR, CEMP, and Black/White, Hispanic/White, and Other/White CEMP ratios for each of our four periods. The Appendix provides 95% CIs for the CEMP ratios.

As expected, both COVID-MR and non-COVID NMR increase monotonically with age in all time periods, for all race/ethnicity groups. CEMP, which adjusts for the higher mortality rates of older individuals from other natural causes, shows a different, more complex pattern. CEMP rises with age, but much more slowly than COVID-MR. CEMP peaks for ages 80–89 for Whites; ages 70–79 for Blacks and Other; and ages 50–59 for Hispanics.

MR levels for Hispanics are substantially greater than for Whites. Yet Hispanic non-Covid NMR is well below that for Blacks, and often below that for Whites, consistent with the known Hispanic life-expectancy advantage over Whites [23]. Hispanic CEMP levels, which reflect high COVID-MR but lower non-Covid NMRs, are far above White levels, and exceed 70% for ages 50–69. Thus, during this period, COVID-19 resulted in an over 70% increase in natural

**Table 1. Summary statistics on study population.**

| | COVID-19 Deaths | Non-COVID Natural Deaths | Population |
|---|---|---|---|
| | N (%) | N (%) | N (%) |
| **Female** | 27,598 (45.26%) | 215,123 (50.09%) | 9,995,462 (51.21% |
| **Male** | 33,381 (54.74%) | 214,348 (49.91%) | 9,521,779 (48.79%) |
| **Race/Ethnicity** | | | |
| White | 46,092 (75.59%) | 360,467 (83.93%) | 14,235,294 (72.94%) |
| Hispanic | 5,538 (9.08%) | 13,924 (3.24%) | 2,038,469 (10.44%) |
| Black | 7,737 (12.69%) | 47,148 (10.98%) | 2,070,258 (10.61%) |
| Other (incl. Asian) | 1,612 (2.64%) | 7,932 (1.85%) | 1,173,220 (6.01%) |
| **Age** | | | |
| 18–39 | 1,114 (1.83%) | 7,852 (1.83%) | 7,353,458 (37.68%) |
| 40–49 | 2,152 (3.53%) | 12,473 (2.90%) | 3,199,776 (16.39%) |
| 50–59 | 5,171 (8.48%) | 34,341 (8.00%) | 3,303,867 (16.93%) |
| 60–69 | 10,955 (17.97%) | 75,721 (17.63%) | 2,857,526 (14.64%) |
| 70–79 | 15,580 (25.55%) | 103,807 (24.17%) | 1,753,745 (8.99%) |
| 80–89 | 16,177 (26.53%) | 115,780 (26.96%) | 830,350 (4.25%) |
| 90+ | 9,830 (16.12%) | 79,484 (18.51%) | 218,519 (1.12%) |
| **Zip-SES (1 = highest)** | | | |
| Quartile 1 | 14,394 (23.66%) | 117,033 (27.30%) | 4,109,751 (21.08%) |
| Quartile 2 | 15,467 (25.42%) | 111,161 (25.93%) | 4,291,941 (22.02%) |
| Quartile 3 | 15,518 (25.50%) | 106,547 (24.86%) | 5,667,031 (29.07%) |
| Quartile 4 | 15,464 (25.42%) | 93,921 (21.91%) | 5,422,880 (27.82%) |
| **Education** | | | |
| Unknown | 1,107 (1.82%) | 6,238 (1.45%) | |
| Not high school grad | 11,554 (18.95%) | 66,011 (15.37%) | 1,669,974 (9.75%) |
| High school grad | 29,226 (47.93%) | 209,488 (48.78%) | 4,908,088 (28.64%) |
| Some college | 10,948 (17.95%) | 79,180 (18.44%) | 5,032,283 (29.37%) |
| College grad or higher | 8,144 (13.36%) | 68,554 (15.96%) | 5,524,618 (32.24%) |

Table shows summary population statistics (from ACS, as of 2020) and mortality statistics (from death certificates) for Indiana, Wisconsin, and Illinois residents aged 18 + in 2020, and decedents over April 1, 2020-June 30, 2022, for Wisconsin and Indiana and April 1, 2020 –December 31, 2021 for Illinois. Racial/ethnic categories are non-Hispanic White, non-Black Hispanic, Black, and Other. Zip-SES is measured using Graham Social Deprivation Index; quartile 1 = highest SES; quartiles are defined using all natural deaths in both states. Non-COVID natural deaths are all deaths excluding manner of death = accident, homicide, or suicide.

mortality for Hispanics in this age range. Excess natural mortality exceeded 60% for Hispanics ages 40–79. Hispanic excess mortality rates are much higher than for any other racial/ethnic group for all ages other than 90+. For ages 18–49, the Hispanic/White CEMP ratio averages over 10:1. For example, for ages 18–39, COVID-19 caused a 50.6% increase in Hispanic natural mortality rates, versus a 4.9% increase for Whites. Hispanic/White CEMP ratios fall with age above age 60, but remain well above those for the other racial and ethnic minority groups.

The patterns for Blacks are quite different than those for Hispanics. Within age bands, CEMP levels for Blacks are consistently above those for Whites. COVID-MR is much higher for Blacks than Whites in all age groups. CEMP attenuates the MR-based differences, because the denominator reflects pre-existing Black-White disparities in non-Covid NMR. Nonetheless, the Black/White CEMP ratio is generally around 2 for persons aged 18–59; it then falls with age for ages 60+.

The Other group, which is numerically dominated by Asians, has CEMP ratios to White well above one, which are higher for ages 18–59 versus 60+ (3.30 vs. 1.82). These reflect

somewhat higher COVID-MRs, coupled with much lower background mortality rates. For ages 70–79, for example, the COVID-MRs for White and Other are nearly the same (0.373% vs. 0.371%), yet the Other/White CEMP ratio is 2.06:1, reflecting lower Non-Covid NMR for the Other group.

The bottom rows in each panel of Table 2 provide a summary measure of racial/ethnic differences in COVID-MR and CEMP levels averaged across broader age ranges, and for all ages. The overall CEMP ratios are 3.70 for Hispanics/Whites; 1.86 for Other/White, and 1.35 for Blacks/Whites. While well above 1, these summary ratios obscure the much higher ratios for

**Table 2. Mortality and CEMP rates and CEMP ratios by age and race/ethnicity for Wisconsin, Indiana, and Illinois.**

**Panel A. Pre-Vaccine Period (April-December 2020). Indiana, Wisconsin, and Illinois**

| Age | White | | | Black | | | | Hispanic | | | | Other | | | |
|---|---|---|---|---|---|---|---|---|---|---|---|---|---|---|---|
| | COVID MR | Non-Covid NMR | CEMP | COVID MR | Non-Covid NMR | CEMP | CEMP Ratio to White | COVID MR | Non-Covid NMR | CEMP | CEMP Ratio to White | COVID MR | Non-Covid NMR | CEMP | CEMP Ratio to White |
| 18–39 | 0.002% | 0.033% | 4.87% | 0.011% | 0.077% | 14.71% | **3.02**\*\*\* | 0.012% | 0.025% | 50.57% | **10.38**\*\*\* | 0.004% | 0.018% | 20.95% | **4.30**\*\*\* |
| 40–49 | 0.009% | 0.130% | 6.98% | 0.051% | 0.322% | 15.80% | **2.26**\*\*\* | 0.067% | 0.107% | 62.73% | **8.99**\*\*\* | 0.015% | 0.061% | 25.00% | **3.58**\*\*\* |
| 50–59 | 0.029% | 0.387% | 7.61% | 0.132% | 0.738% | 17.88% | **2.35**\*\*\* | 0.198% | 0.251% | 78.81% | **10.35**\*\*\* | 0.044% | 0.183% | 24.07% | **3.16**\*\*\* |
| 60–69 | 0.112% | 0.951% | 11.79% | 0.378% | 1.848% | 20.46% | **1.74**\*\*\* | 0.500% | 0.656% | 76.18% | **6.46**\*\*\* | 0.149% | 0.472% | 31.49% | **2.67**\*\*\* |
| 70–79 | 0.373% | 2.187% | 17.06% | 0.795% | 3.074% | 25.86% | **1.52**\*\*\* | 0.960% | 1.434% | 66.96% | **3.92**\*\*\* | 0.371% | 1.057% | 35.09% | **2.06**\*\*\* |
| 80–89 | 1.051% | 5.381% | 19.54% | 1.539% | 6.245% | 24.64% | **1.26**\*\*\* | 1.753% | 3.928% | 44.62% | **2.28**\*\*\* | 0.936% | 3.066% | 30.52% | **1.56**\*\*\* |
| 90+ | 2.841% | 14.802% | 19.20% | 2.914% | 12.340% | 23.61% | **1.23**\*\*\* | 2.892% | 8.584% | 33.69% | **1.76**\*\*\* | 2.700% | 9.521% | 28.36% | **1.48**\*\*\* |
| 18–59 | 0.011% | 0.149% | 7.18% | 0.045% | 0.270% | 16.82% | **2.34**\*\*\* | 0.055% | 0.081% | 68.69% | **9.57**\*\*\* | 0.013% | 0.055% | 23.66% | **3.30**\*\*\* |
| 60+ | 0.449% | 2.580% | 17.39% | 0.728% | 3.097% | 23.52% | **1.35**\*\*\* | 0.826% | 1.435% | 57.57% | **3.31**\*\*\* | 0.367% | 1.159% | 31.68% | **1.82**\*\*\* |
| Total | 0.154% | 0.946% | 16.31% | 0.204% | 0.928% | 22.03% | **1.35**\*\*\* | 0.164% | 0.271% | 60.42% | **3.70**\*\*\* | 0.081% | 0.267% | 30.34% | **1.86**\*\*\* |

**Panel B. Early Vaccine Period (January-June 2021). Indiana, Wisconsin, and Illinois**

| Age | White | | | Black | | | | Hispanic | | | | Other | | | |
|---|---|---|---|---|---|---|---|---|---|---|---|---|---|---|---|
| | COVID MR | Non-Covid NMR | CEMP | COVID MR | Non-Covid NMR | CEMP | CEMP Ratio to White | COVID MR | Non-Covid NMR | CEMP | CEMP Ratio to White | COVID MR | Non-Covid NMR | CEMP | CEMP Ratio to White |
| 18–39 | 0.001% | 0.022% | 5.85% | 0.006% | 0.048% | 11.49% | **1.96**\*\*\* | 0.004% | 0.017% | 23.76% | **4.06**\*\*\* | 0.001% | 0.011% | 9.52% | 1.63 |
| 40–49 | 0.006% | 0.087% | 7.01% | 0.026% | 0.193% | 13.62% | **1.94**\*\*\* | 0.026% | 0.065% | 39.92% | **5.69**\*\*\* | 0.007% | 0.037% | 18.42% | **2.63**\*\*\* |
| 50–59 | 0.020% | 0.243% | 8.14% | 0.060% | 0.477% | 12.65% | **1.55**\*\*\* | 0.067% | 0.169% | 39.38% | **4.84**\*\*\* | 0.024% | 0.124% | 19.50% | **2.40**\*\*\* |
| 60–69 | 0.054% | 0.623% | 8.66% | 0.135% | 1.147% | 11.81% | **1.36**\*\*\* | 0.177% | 0.431% | 41.05% | **4.74**\*\*\* | 0.052% | 0.295% | 17.73% | **2.05**\*\*\* |
| 70–79 | 0.149% | 1.460% | 10.24% | 0.253% | 1.991% | 12.73% | **1.24**\*\*\* | 0.327% | 1.002% | 32.62% | **3.19**\*\*\* | 0.135% | 0.645% | 20.91% | **2.04**\*\*\* |
| 80–89 | 0.317% | 3.400% | 9.32% | 0.427% | 3.753% | 11.37% | **1.22**\*\* | 0.491% | 2.533% | 19.38% | **2.08**\*\*\* | 0.243% | 2.069% | 11.76% | 1.26 |
| 90+ | 0.709% | 8.794% | 8.06% | 0.732% | 7.726% | 9.47% | 1.18 | 0.710% | 5.534% | 12.83% | **1.59**\*\* | 0.629% | 5.684% | 11.07% | 1.37 |
| 18–59 | 0.007% | 0.095% | 7.63% | 0.022% | 0.170% | 12.70% | **1.66**\*\*\* | 0.019% | 0.053% | 36.49% | **4.78**\*\*\* | 0.006% | 0.036% | 17.40% | **2.28**\*\*\* |

*(Continued)*

**Table 2.** (Continued)

| Age | White COVID MR | Non-Covid NMR | CEMP | Black COVID MR | Non-Covid NMR | CEMP | CEMP Ratio to White | Hispanic COVID MR | Non-Covid NMR | CEMP | CEMP Ratio to White | Other COVID MR | Non-Covid NMR | CEMP | CEMP Ratio to White |
|---|---|---|---|---|---|---|---|---|---|---|---|---|---|---|---|
| 60+ | 0.151% | 1.643% | 9.18% | 0.226% | 1.934% | 11.69% | **1.27***** | 0.267% | 0.952% | 28.02% | **3.05***** | 0.113% | 0.731% | 15.52% | **1.69***** |
| Total | 0.054% | 0.603% | 9.01% | 0.069% | 0.581% | 11.92% | **1.32***** | 0.054% | 0.179% | 30.18% | **3.35***** | 0.027% | 0.169% | 15.85% | **1.76***** |

**Panel C. Delta Period (July-December 2021). Indiana, Wisconsin, and Illinois**

| | White | | | Black | | | | Hispanic | | | | Other | | | |
|---|---|---|---|---|---|---|---|---|---|---|---|---|---|---|---|
| Age | COVID MR | Non-Covid NMR | CEMP | COVID MR | Non-Covid NMR | CEMP | CEMP Ratio to White | COVID MR | Non-Covid NMR | CEMP | CEMP Ratio to White | COVID MR | Non-Covid NMR | CEMP | CEMP Ratio to White |
| 18–39 | 0.006% | 0.022% | 26.00% | 0.014% | 0.055% | 26.01% | 1.00 | 0.008% | 0.018% | 47.09% | **1.81***** | 0.004% | 0.013% | 31.17% | 1.20 |
| 40–49 | 0.026% | 0.093% | 28.35% | 0.047% | 0.210% | 22.53% | **0.79*** | 0.036% | 0.072% | 49.83% | **1.76***** | 0.012% | 0.042% | 27.91% | 0.98 |
| 50–59 | 0.057% | 0.255% | 22.45% | 0.086% | 0.488% | 17.61% | **0.78***** | 0.060% | 0.168% | 35.76% | **1.59***** | 0.017% | 0.124% | 13.93% | **0.62***** |
| 60–69 | 0.115% | 0.659% | 17.45% | 0.165% | 1.212% | 13.60% | **0.78***** | 0.121% | 0.441% | 27.49% | **1.58***** | 0.042% | 0.283% | 14.70% | 0.84 |
| 70–79 | 0.212% | 1.565% | 13.53% | 0.237% | 1.967% | 12.07% | **0.89*** | 0.186% | 1.010% | 18.43% | **1.36***** | 0.092% | 0.668% | 13.72% | 1.01 |
| 80–89 | 0.367% | 3.627% | 10.11% | 0.365% | 3.938% | 9.28% | 0.92 | 0.328% | 2.569% | 12.78% | **1.26*** | 0.160% | 2.047% | 7.81% | 0.77 |
| 90+ | 0.651% | 9.526% | 6.83% | 0.593% | 7.792% | 7.61% | 1.11 | 0.552% | 5.968% | 9.25% | 1.35 | 0.386% | 5.725% | 6.74% | 0.99 |
| 18–59 | 0.024% | 0.100% | 24.13% | 0.037% | 0.180% | 20.32% | **0.84***** | 0.023% | 0.055% | 42.23% | **1.75***** | 0.008% | 0.038% | 20.88% | 0.87 |
| 60+ | 0.206% | 1.758% | 11.69% | 0.225% | 1.987% | 11.33% | 0.97 | 0.173% | 0.975% | 17.75% | **1.52***** | 0.079% | 0.731% | 10.80% | 0.92 |
| Total | 0.084% | 0.644% | 12.99% | 0.081% | 0.601% | 13.40% | 1.03 | 0.044% | 0.184% | 24.01% | **1.85***** | 0.022% | 0.171% | 12.62% | 0.97 |

**Panel D. Omicron Period (January-June 2022). Indiana and Wisconsin**

| | White | | | Black | | | | Hispanic | | | | Other | | | |
|---|---|---|---|---|---|---|---|---|---|---|---|---|---|---|---|
| Age | COVID MR | Non-Covid NMR | CEMP | COVID MR | Non-Covid NMR | CEMP | CEMP Ratio to White | COVID MR | Non-Covid NMR | CEMP | CEMP Ratio to White | COVID MR | Non-Covid NMR | CEMP | CEMP Ratio to White |
| 18–39 | 0.003% | 0.022% | 13.50% | 0.004% | 0.042% | 10.00% | 0.74 | 0.005% | 0.019% | 28.57% | **2.12*** | 0.003% | 0.011% | 25.93% | 1.92 |
| 40–49 | 0.011% | 0.085% | 12.98% | 0.027% | 0.180% | 15.00% | 1.16 | 0.024% | 0.088% | 27.08% | **2.09**** | 0.015% | 0.072% | 20.75% | 1.60 |
| 50–59 | 0.032% | 0.228% | 14.22% | 0.047% | 0.419% | 11.13% | 0.78 | 0.031% | 0.172% | 17.74% | 1.25 | 0.030% | 0.216% | 13.82% | 0.97 |
| 60–69 | 0.073% | 0.655% | 11.11% | 0.165% | 1.071% | 15.43% | **1.39***** | 0.117% | 0.453% | 25.82% | **2.32***** | 0.050% | 0.389% | 12.96% | 1.17 |
| 70–79 | 0.165% | 1.553% | 10.63% | 0.244% | 2.105% | 11.60% | 1.09 | 0.302% | 1.099% | 27.51% | **2.59***** | 0.125% | 0.829% | 15.05% | 1.42 |
| 80–89 | 0.310% | 3.567% | 8.69% | 0.502% | 3.743% | 13.42% | **1.54***** | 0.478% | 3.035% | 15.74% | **1.81**** | 0.178% | 2.547% | 6.99% | 0.80 |
| 90+ | 0.630% | 9.217% | 6.84% | 0.861% | 6.703% | 12.84% | **1.88***** | 0.786% | 5.865% | 13.40% | **1.96*** | 0.404% | 6.785% | 5.95% | 0.87 |
| 18–59 | 0.013% | 0.090% | 13.86% | 0.017% | 0.146% | 11.97% | 0.86 | 0.013% | 0.057% | 23.19% | **1.67***** | 0.009% | 0.054% | 17.24% | 1.24 |
| 60+ | 0.159% | 1.707% | 9.31% | 0.248% | 1.845% | 13.42% | **1.44***** | 0.218% | 1.014% | 21.50% | **2.31***** | 0.092% | 0.852% | 10.84% | 1.16 |
| Total | 0.060% | 0.615% | 9.76% | 0.065% | 0.495% | 13.08% | **1.34***** | 0.038% | 0.172% | 22.00% | **2.25***** | 0.023% | 0.182% | 12.42% | **1.27*** |

COVID-19 MR, Non-COVID natural mortality rate, and CEMP for Wisconsin, Indiana, and Illinois during indicated time periods. The Illinois data ends December 31, 2021. The last column in each panel for Black, Hispanic, and Other reports the ratio of CEMP to the corresponding CEMP for White. See Table App-4 in S1 File for confidence intervals for all reported CEMP Ratio to White values. All panels. Data is for decedents so results are effectively weighted by mortality, not population. *, **, *** indicates p < .05, .01, and .001, respectively; significant results (at p < .05 or better) in **boldface**.

COVID MR $= \frac{COVID\ deaths}{Population}$; Non-COVID Natural Mortality Rate (NMR) $= \frac{Non-Covid\ natural\ deaths}{Population}$; CEMP $= \frac{COVID\ deaths}{Non-COVID\ natural\ deaths}$.

the non-elderly. Table App-10 in S1 File provides CEMP ratios that compare minority groups, for example Hispanic/Black.

Fig 1A to 1D provide scatterplots of COVID mortality rates versus Non-Covid natural mortality rates for each of our time periods, expressed as deaths per 100,000 population. The figure shows data points for groups defined by state * age (18–39, 40–49, 50–59, 60–64, 65–69, 70–74, 75–79, 80–84, 85–89, 90–94, 95+) * gender * race/ethnicity (168 groups). Figure App-1 in S1 File provides alternative scatter plots of the number of Covid-19 *deaths* versus non-Covid natural *deaths* for finer groups also defined by zip-SES quintiles. Each plot includes a best-fit regression line. The figure shows the Pearson correlation coefficient between COVID and non-COVID mortality rates, which ranges from 0.82–0.90 depending on the period. Hollow circles indicate data for Whites; blue triangles for Hispanics; orange squares for Blacks, and hollow diamonds for Other. The high correlations confirm that Non-Covid-NMR is strongly associated with COVID-MR. The regression slope in each panel indicates the average CEMP for the full sample. Hispanics are strong outliers, with data points, and thus CEMP, well above the regression line. Blacks and Other also have CEMP values above one, reflected in data points above the regression line, but less strongly so than Hispanics. Table App-11 in S1 File shows correlation coefficients within each racial/ethnic group, which vary from 0.71 (Hispanics in the Delta Period) to 0.99 (Blacks in the Pre-Vaccine Period).

Fig 2A to 2D display CEMP levels graphically for our study periods. In the pre-vaccine period (Fig 2A and 2B), CEMP levels are highest for Hispanics and lowest for Whites, generally rise moderately with age for Whites and Blacks, but for Hispanics are highest for the middle-aged. Fig 2, Panel A, reports CEMP ratios. Overall, the broad pre-vaccine pattern is of higher minority than White CEMP levels, especially for younger ages, and for Hispanics.

In Fig 3, instead of reporting levels, we report ratios of minority group mortality to White mortality. Fig 3A reports CEMP ratios, Fig 3B reports non-Covid-NMR ratios, and Fig 3C reports MR ratios, in each case for Hispanic/White, Black/White, and Other/White COVID-19 mortality. Note that the COVID-MR ratios in Panel C are the product of the CEMP ratios in Panel A times the Non-Covid-NMR ratios in Panel B. The Hispanic/White MR ratios are somewhat lower than CEMP ratios, but the Black/White MR ratios are much higher than the corresponding CEMP ratios and exceed 5:1 in the pre-vaccine period for ages up to 49. The higher Black/White MR ratios reflect a combination of higher background mortality risk and higher Covid-specific risk. These ratios decline sharply with age, reflecting Non-Covid NMRs that also decline sharply with age. For Hispanics and Asians, in contrast, CEMP ratios exceed MR ratios, reflecting the impact of their lower background mortality rates.

## Early vaccine period (1H 2021)

We turn next to CEMP levels and racial/ethnic disparities in CEMP and COVID-MR ratios, in the first half of 2021. During this period, vaccines were initially available to the elderly, healthcare workers, and first responders, but became widely available in April 2021. We present numerical results in Table 2, Panel B and graphical results Figs 1–3. For Whites, the CEMP age pattern is very different than in the pre-vaccine period. CEMP now rises more gently with age, and peaks for ages 70–79, even though COVID-MR still rises strongly with age.

For Hispanics, the very high non-elderly Hispanic/White CEMP ratios seen in the pre-vaccine period drop, but remain much higher than for Blacks or Other; these ratios are over 4:1 for ages 18–69, but drop at higher ages. In all time periods, Hispanic MR ratios to White are somewhat below CEMP ratios, reflecting lower Hispanic background mortality rates. In Fig 1, the data points for Hispanics are well above the regression line, but less so than in the pre-vaccine period.

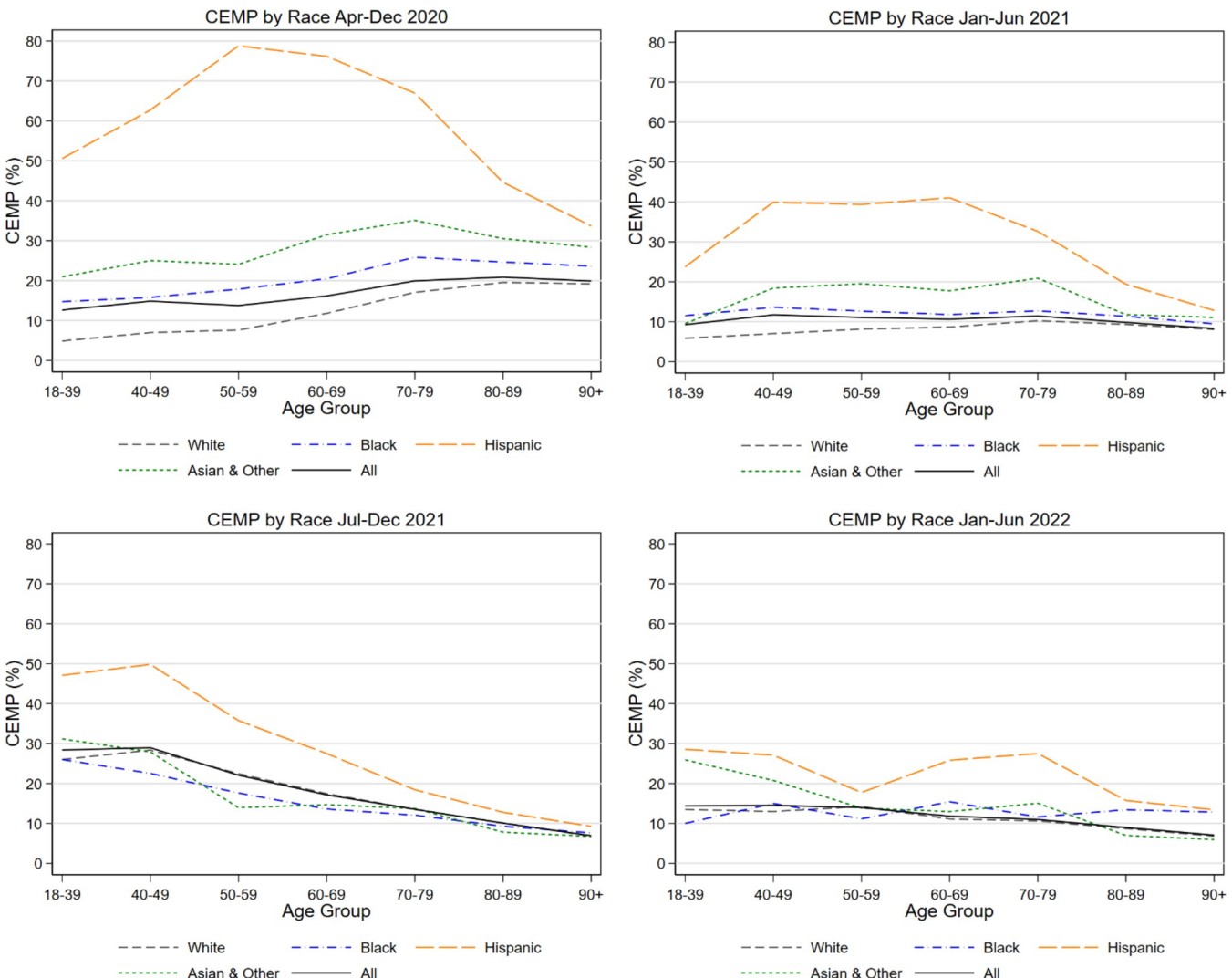

**Fig 2. COVID Excess Mortality Percentage (CEMP) levels by gender and race/ethnicity.** Figure shows, CEMP for White, Hispanic, Black, and Other by age range for all Indiana, Wisconsin, and Illinois adult decedents (age 18+) from natural causes, for indicated time periods. Illinois data covers only the first three periods (through year-end 2021).

For Blacks, CEMP ratios to White during the early vaccine period are close to 2:1 for ages 18–49; these ratios decline at higher ages but remain above 1 for all ages. In all time periods, Black MR ratios to White are 2–2.5 times higher than CEMP ratios, reflecting higher Black background mortality rates. Thus, Black MR ratios to White remain at or above 3:1 through age 59.

For the Other group, CEMP ratios are generally around 2–2.5:1 for ages 18–79, but drop at higher ages. In all time periods, Other MR ratios to White are well below CEMP ratios, reflecting lower background mortality rates. In the early-vaccine and later periods, Other MR ratios to White are generally at or below 1:1.

## Delta period (Second Half of 2021)

In the Delta period, primary vaccination was available for all ages. Booster rollout began in October 2021, initially for ages 60+, healthcare workers, and other high-risk persons. White

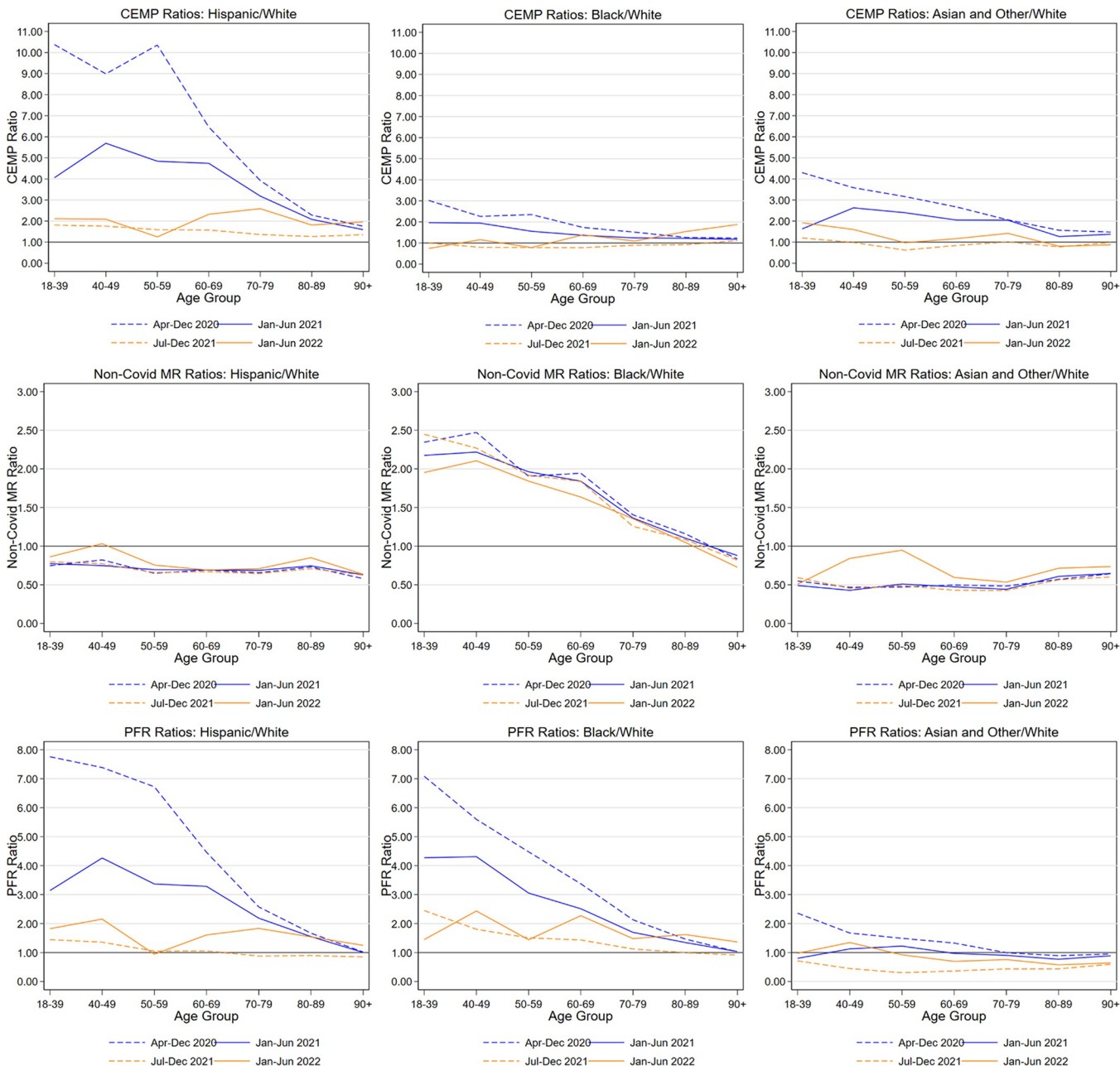

**Fig 3. COVID excess mortality percentage and mortality rate ratios by race/ethnicity and age group.** Figure shows, for Indiana, Wisconsin, and Illinois natural-cause adult decedents (age 18+), CEMP ratios (**Panel A**), Non-Covid MR ratios (**Panel B**), and MR ratios (**Panel C**), and for Hispanic, Black, and Other to White, by age range for indicated time periods. Illinois data covers only the first three periods (through year-end 2021). **Panel A. CEMP Ratios: Hispanic/White; Black/White; and Other/White. Panel B. Non-Covid MR Ratios: Hispanic/White; Black/White; and Other/White. Panel C. MR Ratios: Hispanic/White; Black/White; and Other/White.**

CEMP ratios are now highest for persons aged 18–49 but then decline with age, consistent with the increase in uptake of primary and booster vaccination with age [15], and likely also with older persons doing more than younger persons to reduce infection risk. For Whites, the CEMP age pattern is very different than in the pre-vaccine period. CEMP now rises more gently with age, and peaks for ages 70–79, even though COVID-MR still rises strongly with age.

For Hispanics, CEMP ratios to White during the Delta Period drop again, and are roughly 1.6 to 1.8:1 for ages 18–69. In the scatterplot in Fig 1, the Hispanic data points are generally above the best-fit line, but much less so than in earlier period.

For Blacks, CEMP ratios to White are generally at or below 1:1 for most age groups, and are significantly below 1:1 for Blacks ages 18–59. Yet Black/White COVID-MR ratios remain well above 1:1, reflecting higher Black background mortality risk.

For Other, CEMP ratios to White are generally at or below 1:1 for most age groups. MR ratios to White are even lower, reflecting lower Other background mortality risk.

## Omicron period (2H 2021)

In the Omicron period, boosters were fully available to all ages, although uptake was limited, especially for younger persons [15]. For Whites, CEMP levels are broadly similar for ages 18–59, and decline at higher ages.

For Hispanics, CEMP ratios to White rise during the Omicron Period, relative to the Delta Period, and are generally in the 2–2.5:1 range for ages 18–79. For Blacks, CEMP ratios to White are broadly similar for ages 18–59, but rise for older ages relative to the Delta Period. For Other, CEMP ratios to White also generally rise, relative to the Delta Period.

## Multivariate logistic regression results

The CEMP measure controls for population health through the denominator. Thus, minority/White ratios substantially different than one reflect primarily Covid-specific factors, rather than underlying health differences. In Table 3, we use multivariate logistic regression to assess whether controlling for additional factors, beyond those in Table 2, materially changes the minority/White CEMP ratios reported above. We run separate regressions for each time period, for ages 18–59 and ages 60+. The predictors are race/ethnicity (White is omitted); age groups (youngest group is omitted), zip-SES quartiles (highest quartile is omitted), education (college-graduate-or-higher is omitted), and marital status (unknown is omitted). CIs are in brackets.

The odds ratios for Black, Hispanic, and Other reported in Table 3 correspond to the CEMP ratios in Table 2. Controlling for the additional predictors only modestly changes coefficient magnitudes. For example, the Hispanic/White odds ratio of 8.82 for ages 18–59 in the pre-vaccine period, is somewhat below the 9.57 Hispanic/White CEMP ratio in Table 2. For this age group and time period, the Black/White odds ratio of 2.21 is close to the Black/White CEMP ratio of 2.34; and the Other/White odds ratio of 3.19 is close to the Other/White CEMP ratio of 3.30. Thus, zip-SES, education, and marital status explain only a small part of the CEMP-based disparities reported above.

**Validity of the CEMP measure.** We assess the validity of the CEMP measure in several ways. First, we confirm that natural mortality in 2019 is strongly associated with COVID-19 mortality in 2020 (pre-vaccine period) (Appendix Figure App-2 in S1 File) and that COVID-19 mortality during the pandemic period is strongly associated with contemporaneous non-COVID mortality (Fig 1). Second, we confirm that COVID-19 deaths are not materially undercounted in Indiana and Wisconsin, and only modestly undercounted in Illinois, by comparing measured Non-COVID natural deaths in each state during the pandemic period to predicted natural deaths based on extrapolation from 2017–2019 (Appendix Figure App-3 in S1 File).

Third, we study an alternative CEMP measure, in which we divide COVID-19 mortality by natural mortality for the corresponding months in *2019* instead of contemporaneous non-COVID natural mortality; we call this measure CEMP-2019. See Appendix Table App-8 in

**Table 3. Racial/Ethnic disparities in COVID mortality rates: Multivariate logit analysis.**

| | 18–59 | | | | 60+ | | | |
|---|---|---|---|---|---|---|---|---|
| Category | Apr-Dec 2020 | Jan-Jun 2021 | Jul-Dec 2021 | Jan-Jun 2022 | Apr-Dec 2020 | Jan-Jun 2021 | Jul-Dec 2021 | Jan-Jun 2022 |
| Black | **2.21***** | **1.79***** | **0.84**** | 0.96 | **1.33***** | **1.22***** | **0.83***** | **1.28***** |
| Hispanic | **8.86***** | **5.19***** | **1.80***** | **1.80***** | **3.26***** | **2.97***** | **1.43***** | **2.07***** |
| Other | **3.24***** | **2.36***** | 0.87 | 1.29 | **1.82***** | **1.75***** | 0.92 | 1.15 |
| Male | **1.14**** | 1.11 | **1.16***** | 1.01 | **1.20***** | **1.27***** | **1.15***** | **1.23***** |
| Age range | | | | | | | | |
| 40–49 | **1.21*** | 1.32 | 1.00 | 1.00 | | | | |
| 50–59 | **1.38***** | **1.38***** | **0.78***** | 0.99 | | | | |
| 70–79 | | | | | **1.32***** | **1.14***** | **0.81***** | 0.97 |
| 80–89 | | | | | **1.45***** | 1.03 | **0.61***** | **0.81***** |
| 90+ | | | | | **1.46***** | **0.92*** | **0.43***** | **0.66***** |
| zip SES Quartile | | | | | | | | |
| 2 | 1.05 | 1.07 | 0.97 | 0.89 | **1.13***** | **1.11**** | **1.19***** | 1.06 |
| 3 | 1.15 | 1.16 | 1.00 | 0.99 | **1.16***** | **1.16***** | **1.19***** | **1.10**** |
| 4 | **1.29***** | 0.97 | 0.92 | **0.75*** | **1.22***** | **1.17***** | **1.16***** | **1.16***** |
| Education Level | | | | | | | | |
| Unknown | 1.08 | 1.08 | 0.93 | 0.82 | **1.44***** | 0.96 | 1.14 | 1.06 |
| No High School | **0.77***** | **0.73**** | 0.99 | 0.82 | **1.09***** | **1.10*** | **1.48***** | **1.38***** |
| High School | **0.83**** | 0.96 | **1.39***** | 0.94 | 1.02 | **1.12***** | **1.28***** | **1.29***** |
| Associate/Some College | **0.83**** | 1.18 | **1.47***** | 1.17 | 0.99 | **1.08*** | **1.22***** | **1.23***** |

Table reports odds ratios for indicated variables calculated using multivariate logit models of the probability of COVID death among the sample of natural deaths, and 95% confidence intervals (CIs). Sample period from April 1, 2020, through June 30, 2022. Sample includes natural cause deaths for persons 18 and older occurring in the states of Wisconsin, Indiana, and Illinois. Data for Illinois covers only the first three periods. Age ranges for ages 18–59 are 18–39 (omitted), 40–49 (bin 2), and 50–59 (bin 3). Age ranges for age 60+ are 60–69 (omitted) 70–79 (bin 2), 80–89 (bin 3) and 90+ (bin 4). Omitted categories for the other variables are: White (race/ethnicity), Female, First (most affluent) zip-SES quartile, and College degree or higher education. *, **, *** indicates p < .05, .01, and .001, respectively; significant results (at p < .05 or better) in boldface.

S1 File. This avoids the risk of bias in the CEMP measure due to undercounting COVID-19 deaths during the pandemic period, which would both lower the CEMP numerator and increase the denominator. CEMP-2019 values are close to, but generally modestly lower than the corresponding CEMP values reported in the text. Minority/White CEMP-2019 ratios are similar to the corresponding CEMP ratios reported in the text (Appendix Table App-9 in S1 File compares the two). These results are consistent with the modest undercounting of COVID-19 deaths in Illinois, noted above.

**Generalizability to other states.** In related work, we report national results for 2020, which are consistent with the results reported below for the pre-vaccine period, including very high Hispanic/White CEMP ratios [16].

## Discussion

### Overview

We report results for COVID-19 mortality risk using a new measure–the COVID Excess Mortality Percentage (CEMP), which adjusts COVID-19 mortality risk for population health. (We have used and validated this measure in related work) [16–18,24]. Comparing disparities across racial/ethnic groups measured using CEMP as the outcome to those measured using COVID-MR lets us separate COVID-specific effects from those that reflect differences in

underlying health. The disproportionate impact of COVID-19 on Hispanic population reflects very large pandemic-specific factors, partly offset by better background health. In contrast, the disproportionate effect on Blacks reflects a combination of pandemic-specific factors and worse underlying population health.

In 2020, COVID-19 was the largest cause of natural death for Hispanics. For ages through 79, COVID-19 was far ahead of heart disease, cancer, and all other natural causes. While Hispanic/White disparities have been observed before, the large magnitudes that we find have been obscured in studies that use COVID-MR or excess mortality as the outcome of interest, many of which lack good controls for underlying health, and thus for expected mortality assuming similar COVID exposure [13,25,26]. Prior work also often does not control for population age, or else reports age-adjusted mortality rates, rather than providing breakdowns by age group. For example, a recent CDC study reports age-adjusted Hispanic/White COVID-19 mortality rate ratios of 2.78 for 2020 and 1.71 for 2021, far below the levels we find for younger ages [27]. Prior work has also not stressed the much greater disparities for younger persons, nor the especially high COVID-specific burden for younger Hispanics. The Hispanic/White CEMP ratio in the pre-vaccine period is even higher for men (Appendix Table App-3 in S1 File). One possible explanation could be that Hispanics disproportionately work in occupations with high COVID-19 infection risk [28].

During the pre-vaccine period, CEMP ratios were generally highest for the elderly. In contrast, in the early vaccine period, the contribution of COVID-19 to overall natural mortality was highest in middle age ranges; and during the Delta and Omicron periods, CEMP ratios generally fell with age. These differing patterns likely reflect both higher elderly vaccination rates (both primary vaccination, and booster does when they became available) and behavioral differences, with the elderly being more successful at avoiding infection.

## Variation with age and time period

The manner in which CEMP levels vary with age, with time period, and across racial and ethnic groups are complex and defy easy explanation. Systemic factors that affect background health cannot explain the disparities in CEMP levels we report, because the CEMP measure controls for background health. The different patterns in CEMP levels by race/ethnicity, by time period, and by age range within each racial/ethnic group have not, to our knowledge, been previously reported.

## Advantages of the CEMP measure

Any mortality measure will have strengths and limitations. However, CEMP has attractive features and can complement measures based on COVID-MR or excess mortality. Most importantly, it controls for population health, as reflected in non-COVID NMR. Population health is otherwise difficult to observe. Comorbidity data from electronic medical records (EMR) is subject to the variable quality of reporting of comorbid conditions, and in the U.S. is not available at the population level. Thus, most COVID-19 studies that address the impact of comorbid conditions on outcomes examine people who present for medical care for COVID-19 infection [9,29], missing the impact of underlying health factors on who becomes infected and infection severity, or else selected populations (e.g., persons receiving care at Veteran's Administration facilities) [30–32].

CEMP also has important implementation advantages, notably the feasibility of gathering complete population data, since it relies on death certificates, which are available for all decedents. This has advantages over approaches that require estimating the population at risk. Some populations may be undercounted in population statistics because of non-participation

in the Census or provision of inaccurate data. While race/ethnicity can also be inaccurately captured in death-certificate data, we have no reason to expect that any inaccuracies will differ systematically between those who die of COVID-19 versus other natural causes. Death certificates usually have personal identifiers, making it feasible to enhance analyses by adding information on the decedents' COVID-19 vaccination or infection status, which is collected at the individual level by many states. Linkage to individual-level EMR data may also be possible.

Aron and Muellbauer have argued that "excess mortality"–deaths in excess of those predicted based on pre-pandemic experience–is a more complete measure of COVID-19 impact since it does not rely on coding accuracy and includes collateral deaths such as deaths caused by failure to seek care because of fear of contracting COVID-19 during the healthcare encounter [33,34]. But excess mortality requires estimating expected deaths based on pre-pandemic experience, which is difficult for smaller subgroups and less reliable over longer estimation periods. There is also no obvious reason why downward bias in measuring COVID-19 deaths should cause important bias in CEMP or MR ratios. At the same time, it would be possible to study excess mortality in a manner similar to how we use CEMP, using a "P-value" for excess mortality (defined as excess mortality as a percentage of expected mortality) [33,35].

## Implications of our findings regarding COVID-19 impact by age range and racial/ethnic group

While many reports have emphasized the impact of COVID-19 on older adults, our use of CEMP as an outcome demonstrates that COVID-19 was a major contributor to death in middle aged adults throughout the pandemic. Indeed, during the Delta and Omicron periods COVID-19 caused a larger percentage increase in mortality for persons aged 18–59 than for persons aged 60+. Thus, continued attention to mitigation of risk in the working age population is needed.

The Hispanic/White CEMP ratios in the pre-vaccine period are stunning. For ages 18–59, they average 9.57:1 and over 10:1 for men. Only a small part of these ratios is explained by population health, zip-SES, or education. These disparities call for close study of the reasons for them, and why they vary so strongly with age, gender, and between the pre-vaccine and later periods. Any explanation will surely be multifactorial, and will likely depend on factors we don't observe, including infection rates [2,36], (many infections are not captured in the available datasets) [37], racial/ethnic variation in vaccination rates [36], variation across hospitals in COVID-19 survival rates [38], possible variation across racial/ethnic groups in where and how soon they seek treatment, and possible variation in post-hospitalization outcomes [9,20,39,40].

Our CEMP ratio data underscores the very large COVID-19 impact on Hispanic mortality, especially for the non-elderly, which contrasts with the Hispanic paradox [23], for other causes of death, of life-expectancy advantage relative to Whites, despite socioeconomic disadvantage. Research is needed to understand why Hispanic versus White COVID-19 mortality has been so different than mortality from cancer, heart disease and other natural causes.

For Blacks, comparing CEMP and Covid- MR results emphasizes that the impact of COVID-19 on Black mortality reflects a combination of COVID-19 specific factors and worse underlying population health. While understanding COVID-19-specific factors will benefit the Black community, attention to systemic factors that affect overall health is even more vital.

## Implications of vaccination and vaccination rate disparities

Other research reports relatively high uptake of primary vaccination for Hispanics (after a slower start) and Asians, with lower uptake by Whites, and the lowest uptake rates for Blacks

[41]. Our study provides evidence that despite difference in uptake rates and timing, vaccine availability was of particular benefit to minority groups, in reducing disparities in COVID-19 specific risk. Although we cannot show causation with our data, minority/White disparities in CEMP and MR ratios fell during 2021, as vaccines became more widely available. Similarly, we find in prior work on the elderly that the ratio of elderly to middle-aged COVID-19 mortality fell dramatically over January-April 2020, as the elderly because eligible for vaccination, generally before the non-elderly, with the largest effects for ages 85+ [17]. The groups at highest risk without vaccines benefited the most from having vaccines available, even though many people chose to remain unvaccinated.

## Limitations of CEMP and this paper

CEMP and CEMP ratios provide a measure of COVID-19 impact that complements but does not replace more common measures (COVID-MR and excess mortality). CEMP does not address other important COVID-19 health outcomes, including hospitalizations, ICU admissions and long COVID.

Non-Covid natural mortality rates, used in the CEMP denominator, cannot capture all health and other factors that affect individual risk, including behavioral factors. For example, diabetes is an important risk factor for COVID-19 mortality. Minority groups have higher diabetes rates than Whites [42]; these could have contributed to the minority/White disparities we observe. There is evidence that the Hispanic life expectancy advantage is shrinking over time (Appendix Figure App-5 in S1 File), the underlying causes could affect COVID-19 risk before they affect overall mortality. We did not have access to population health measures, except through the limited lens of death certificates.

Moreover, CEMP as a single measure may obscure factors that may warrant individual attention. Zip-SES, education, and marital status were only weakly associated with CEMP levels (Table 3), but stronger associations between these variables and COVID-MR have been found in other studies [6,43].

CEMP will be a downward-biased measure of excess mortality due to COVID for several reasons. First, it ignores COVID's effect on natural deaths, not directly attributable to COVID in mortality records. There is evidence of an increase in cardiovascular mortality following COVID-19 infection [44]. Second, CEMP will be downward biased if COVID-19 deaths are underreported on death certificates. Third, care for other conditions could be disrupted during peak-COVID periods. We find evidence consistent with some undercounting of COVID deaths and/or medical care disruptions in Illinois during peak COVID periods, but no similar evidence for Indiana or Wisconsin. Conversely, CEMP may be an upward biased measure of excess mortality among the old and frail, because some COVID-19 decedents would have died soon from other causes [45].

Any bias in measuring CEMP will affect minority/White ratios only if the bias differs across groups. CEMP-2019 levels should not be affected by these sources of bias, and are somewhat lower than those we report (Table App-8 in S1 File), but minority/White CEMP-2019 ratios are similar to those we report (Table App-9 in S1 File).

We examined only three Midwestern states, but confirm notably high Hispanic/White CEMP ratios nationally for 2020 in related work [16]. We lacked vaccination data. We also lacked sufficient sample size to decompose the broad Other group. That group includes Asians, who as a group have higher vaccination rates than Whites [36,46], lower COVID and non-Covid natural mortality rates [3,16], although large differences in COVID outcomes exist within this broad group [47]; Native Americans, who have faced relatively high COVID-19 infection and mortality rates [6,16,48,49], and other (e.g., mixed race). We also lacked

sufficient sample size to decompose the other broad groups, which are likely to also exhibit substantial intra-group differences.

## Conclusion

We use a new measure of COVID mortality risk (CEMP), which uses Non-COVID natural mortality risk to adjust COVID-19 mortality risk for underlying population health. CEMP ratios for racial and ethnic minorities to Whites were; large, especially in the pre-vaccine period extremely so for younger Hispanics. The minority/White ratios were generally lower for ages 60+, and substantially lower, sometimes at or below 1:1, in the Delta and Omicron periods. Large disparities in CEMP ratios suggest that differences in COVID-19 mortality, particularly between Hispanic and non-Hispanic White communities, are inadequately explained by known disparities in risk factors or healthcare. The large variation in CEMP levels by age, gender, race/ethnicity, and time period (pre-vaccine versus vaccine-available) defies easy explanation.

## Supporting information

**S1 File. Appendix for "Evidence on COVID-19 Mortality and Disparities Using a Novel Measure, COVID Excess Mortality Percentage: Evidence from Indiana, Wisconsin, and Illinois".**
(DOCX)

## Author Contributions

**Conceptualization:** Vladimir Atanasov, Jeff Whittle, John Meurer, Benjamin W. Weston, Bernard Black.

**Data curation:** Natalia Barreto, Lorenzo Franchi, Qian (Eric) Luo, Andy Ye Yuan, Ruohao Zhang, Bernard Black.

**Formal analysis:** Vladimir Atanasov, Natalia Barreto, Andy Ye Yuan, Bernard Black.

**Funding acquisition:** John Meurer, Bernard Black.

**Investigation:** Bernard Black.

**Methodology:** Vladimir Atanasov, Qian (Eric) Luo, Andy Ye Yuan, Bernard Black.

**Project administration:** Jeff Whittle, Bernard Black.

**Supervision:** John Meurer, Bernard Black.

**Validation:** Vladimir Atanasov, Bernard Black.

**Visualization:** Vladimir Atanasov, Andy Ye Yuan, Bernard Black.

**Writing – original draft:** Vladimir Atanasov, Bernard Black.

**Writing – review & editing:** Vladimir Atanasov, Jeff Whittle, John Meurer, Benjamin W. Weston, Andy Ye Yuan, Bernard Black.

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
