## [Decision Letter · Decision Letter 0]

4 Aug 2023

PONE-D-23-10973Evidence on COVID-19 Mortality and Disparities Using a Novel Measure, COVID Excess Mortality Percentage: Evidence from Indiana, Wisconsin, and IllinoisPLOS ONE

Dear Dr. Yuan,

Thank you for submitting your manuscript to PLOS ONE. After careful consideration, we feel that it has merit but does not fully meet PLOS ONE’s publication criteria as it currently stands. Therefore, we invite you to submit a revised version of the manuscript that addresses the points raised during the review process.

The reviewers reached a consensus of major revision. After reading the comments and the submitted manuscript, the editor agrees with such consensus and believes that reviewers' comments are crucial for enhancing the quality of the manuscript to reach the publication standard of PLOS ONE. Please revise your paper thoroughly according to the comments from the two reviewers and provide an itemized response letter.

We look forward to receiving your revised manuscript.

Kind regards,

Chenfeng Xiong

Academic Editor

PLOS ONE

Journal Requirements:

When submitting your revision, we need you to address these additional requirements. 1. Please ensure that your manuscript meets PLOS ONE's style requirements, including those for file naming. The PLOS ONE style templates can be found at https://journals.plos.org/plosone/s/file?id=wjVg/PLOSOne_formatting_sample_main_body.pdf and https://journals.plos.org/plosone/s/file?id=ba62/PLOSOne_formatting_sample_title_authors_affiliations.pdf 2. Thank you for stating in your Funding Statement: "Research reported in this publication was supported by the National Center for Advancing Translational Sciences of the National Institutes of Health under Award Number UL1TR001436 to the Medical College of Wisconsin Clinical and Translational Science Institute of Southeast Wisconsin.  The content is solely the responsibility of the authors and does not necessarily represent the official views of the NIH." Please provide an amended statement that declares *all* the funding or sources of support (whether external or internal to your organization) received during this study, as detailed online in our guide for authors at http://journals.plos.org/plosone/s/submit-now.  Please also include the statement “There was no additional external funding received for this study.” in your updated Funding Statement. Please include your amended Funding Statement within your cover letter. We will change the online submission form on your behalf. 3. Thank you for stating the following financial disclosure: "Research reported in this publication was supported by the National Center for Advancing Translational Sciences of the National Institutes of Health under Award Number UL1TR001436 to the Medical College of Wisconsin Clinical and Translational Science Institute of Southeast Wisconsin.  The content is solely the responsibility of the authors and does not necessarily represent the official views of the NIH." Please state what role the funders took in the study. If the funders had no role, please state: ""The funders had no role in study design, data collection and analysis, decision to publish, or preparation of the manuscript."" If this statement is not correct you must amend it as needed. Please include this amended Role of Funder statement in your cover letter; we will change the online submission form on your behalf. 4. In your Data Availability statement, you have not specified where the minimal data set underlying the results described in your manuscript can be found. PLOS defines a study's minimal data set as the underlying data used to reach the conclusions drawn in the manuscript and any additional data required to replicate the reported study findings in their entirety. All PLOS journals require that the minimal data set be made fully available. For more information about our data policy, please see http://journals.plos.org/plosone/s/data-availability. Upon re-submitting your revised manuscript, please upload your study’s minimal underlying data set as either Supporting Information files or to a stable, public repository and include the relevant URLs, DOIs, or accession numbers within your revised cover letter. For a list of acceptable repositories, please see http://journals.plos.org/plosone/s/data-availability#loc-recommended-repositories. Any potentially identifying patient information must be fully anonymized. Important: If there are ethical or legal restrictions to sharing your data publicly, please explain these restrictions in detail. Please see our guidelines for more information on what we consider unacceptable restrictions to publicly sharing data: http://journals.plos.org/plosone/s/data-availability#loc-unacceptable-data-access-restrictions. Note that it is not acceptable for the authors to be the sole named individuals responsible for ensuring data access. We will update your Data Availability statement to reflect the information you provide in your cover letter. 5. Your ethics statement should only appear in the Methods section of your manuscript. If your ethics statement is written in any section besides the Methods, please delete it from any other section. 

Reviewers' comments:

Reviewer's Responses to Questions

**Comments to the Author**

1. Is the manuscript technically sound, and do the data support the conclusions?

Reviewer #1: Yes

Reviewer #2: Partly

2. Has the statistical analysis been performed appropriately and rigorously? 

Reviewer #1: No

Reviewer #2: Yes

3. Have the authors made all data underlying the findings in their manuscript fully available?

Reviewer #1: No

Reviewer #2: No

4. Is the manuscript presented in an intelligible fashion and written in standard English?

Reviewer #1: Yes

Reviewer #2: Yes

5. Review Comments to the Author

Reviewer #1: This paper examines a new measure—the COVID Excess Mortality Percentage or CEMP—as an alternative way to understand disparities in COVID-19 throughout the course of the pandemic in three midwestern states. The separation into different pandemic periods (pre-vaccine, early vaccine, Delta, Omicron) is a strength of this paper and shows how disparities have evolved over time.

My primary concern is with the use of contemporary non-COVID-19 mortality rates in the CEMP rather than pre-pandemic mortality. The introduction notes that this novel measure takes into account background risk and was validated by comparing 2019 natural mortality to 2020 COVID-19 mortality. That seems very reasonable, but the use of contemporaneous COVID-19 and other natural mortality in the present study does not. Prior research has shown that COVID-19 is a competing risk of death with several natural causes, and may be so differentially by population. The paper would be more convincing if pre-pandemic mortality were used so as not to contaminate the non-COVID mortality rates used for background risk.

That concern aside, the paper needs to better motivate and provide intuition about how to interpret the CEMP measure. I do not understand the “excess” in COVID Excess Mortality Percentage. Because of the way it is defined, it’s not clear what this in excess of.

While the NMR takes into account background health to some extent, it does not fully capture it. For example, several scholars have questioned whether the Hispanic paradox can persist given rising diabetes and obesity rates. In the absence of the pandemic, these underlying risk factors may have taken many years to manifest in increased mortality in the Hispanic population, but were clearly associated with increased risk of COVID-19. The fact that NMR, and thus CEMP, cannot fully capture underlying health should be discussed more in the limitations section.

The methods section mentions that immune-compromised individuals were excluded. What is the rationale for their exclusion? These individuals would certainly have

The use of “COVID-PFR” throughout the manuscript was confusing as there is standard demographic terminology “COVID-19 mortality rate” that is commonly used in the literature.

Figure 1 typo: the title says 2019 NMR but the axis labels show contemporaneous NMR rates.

Why do the scatterplots display absolute numbers of deaths rather than population rates? This makes it hard to see racial/ethnic disparities because minorities are all clumped together near 0 because of low population size.

The text references PFR ratios when discussing Table 2, but these are not displayed so it is hard to follow.

Reviewer #2: The authors studied COVID-19 mortality and disparities in Indiana, Wisconsin, and Illinois. They studied that mortality risk and racial/ethnic disparities in COVID-19 mortality and how both vary with age, gender, and time period. This manuscript uses a novel measure called COVID Excess Mortality Percentage (CEMP) to inform COVID-19 mortality rates and racial/ethnic disparities. CEMP is defined as the COVID-19 mortality rate divided by the non-COVID natural mortality rate, converted as a percentage. By studying the CEMP along with the COVID-19 population fatality rate (COVID-PFR), the authors are able to separate the effects of background health from Covid-specific factors affecting COVID-19 mortality. The authors concluded that CEMP levels for racial and ethnic minority groups, especially for Hispanics, can be very high relative to White levels. Disparities were generally lower after age 60 and declined over the sample period. Socio-economic status and education contribute very little to these disparities.

Generally, I thought that the manuscript was well-written (with a few minor typos) and very completed and detailed in their description, methodology, analysis, discussion, and limitations. I have a few questions regarding the manuscript that I hope that the authors could clarify either through the review process or by updating their paper.

1. Why were minority ethnic groups only compared to the white ethnic group? Why not perform an analysis between different ethnic groups as well?

2. I do not think that saying that the CEMP denominator strongly predicts COVID-19 mortality is valid. You can say that they are strongly correlated, but not necessarily a prediction factor, at least from my point of view. What makes this a prediction? What are some explanations for this correlation/prediction?

3. For the correlation between natural mortality and COVID-19 mortality, the data points in the higher values of natural non-Covid deaths are more spread out than near the origin. This potentially could indicate that this model is not a good predictor for high mortality (since the confidence interval will need to be wider).

4. Why was the correlation between natural mortality and COVID-19 mortality done for the entire data set? Was it done for each ethnic group as well?

5. Additionally, the authors mentioned that all ethic minority groups had the majority of their data points above the regression line. All this means is that their CEMP is higher than White (which make up most of the data points). What is the purpose of this figure as this can be calculated numerically?

Some typos that I caught while reading:

1. Page 7: space required after "=" for Non-COVID NMR

2. Page 7: If you are using equation numbers, please use different equation numbers for different equations (you have equation #1 appearing twice). Additionally, there is no purpose of including equation numbers if you will not be referencing the equation in your text.

3. Page 8: Should be "Other =" instead of "Other -"

4. Page 8, second-last paragraph: should be "First, Hispanics [...]", instead of "First. Hispanics [...]"

5. Page 14: Last paragraph starting with "In 2020", the first second doesn't make sense.

6. Page 15, first paragraph: should be "During the pre-vaccine period" instead of "During the pre-vaccine Period"

7. Page 18, second paragraph: should be "COVID-PFR" instead of "CCOVID-PFR"

6. PLOS authors have the option to publish the peer review history of their article (what does this mean?). If published, this will include your full peer review and any attached files.

Reviewer #1: No

Reviewer #2: No

---

## [Author Response · Author response to Decision Letter 0]

14 Sep 2023

Please see a separate response-to-referee letter

---

## [Decision Letter · Decision Letter 1]

31 Oct 2023

PONE-D-23-10973R1Evidence on COVID-19 Mortality and Disparities Using a Novel Measure, COVID Excess Mortality Percentage: Evidence from Indiana, Wisconsin, and IllinoisPLOS ONE

Dear Dr. Yuan,

Thank you for submitting your manuscript to PLOS ONE. After careful consideration, we feel that it has merit but does not fully meet PLOS ONE’s publication criteria as it currently stands. Therefore, we invite you to submit a revised version of the manuscript that addresses the points raised during the review process.

Please address the remaining comments from Reviewer 1. As this is the second round of review and only minor revision comments were received by reviewers, the editor reached a minor revision decision. And please finalize the paper and resubmit. It is worth noting that both Reviewer 1 and Reviewer 2 brought up seemingly editorial related comments. Please work on editing your paper thoroughly to make sure the writing is clean and accurate.

We look forward to receiving your revised manuscript.

Kind regards,

Chenfeng Xiong

Academic Editor

PLOS ONE

Journal Requirements:

Reviewers' comments:

Reviewer's Responses to Questions

**Comments to the Author**

1. If the authors have adequately addressed your comments raised in a previous round of review and you feel that this manuscript is now acceptable for publication, you may indicate that here to bypass the “Comments to the Author” section, enter your conflict of interest statement in the “Confidential to Editor” section, and submit your "Accept" recommendation.

Reviewer #1: (No Response)

Reviewer #2: All comments have been addressed

2. Is the manuscript technically sound, and do the data support the conclusions?

Reviewer #1: Yes

Reviewer #2: Yes

3. Has the statistical analysis been performed appropriately and rigorously? 

Reviewer #1: Yes

Reviewer #2: Yes

4. Have the authors made all data underlying the findings in their manuscript fully available?

Reviewer #1: Yes

Reviewer #2: Yes

5. Is the manuscript presented in an intelligible fashion and written in standard English?

Reviewer #1: Yes

Reviewer #2: Yes

6. Review Comments to the Author

Reviewer #1: The authors’ explanation of the interpretation of CEMP makes more sense in this revision, but CEMP is rarely interpreted in the results section, which limits the readability and interpretability of the findings for readers. For example, on page 12 the authors point out that CEMP levels exceed 70% for ages 50-59 but do not interpret that this means that there are 70% more natural deaths than would be expected due to COVID for this age group (under the assumption that COVID deaths occurred only among those who otherwise would not have died).

The text states that “For Illinois, there is evidence of some undercounting of COVID-19 deaths: natural-non-COVID deaths rise during peak COVID-mortality periods (Figure App-4, Panel C)” and notes that this implies CEMP will be underestimated. There is evidence that mortality rates from several causes of death increased due to medical disruptions during the pandemic, so how can we be sure that these peaks imply undercounts of COVID-19 and not indirect effects of the pandemic? How does a real change in non-COVID natural mortality that does not reflect undercounting impact the validity of the CEMP measure?

I appreciate that the authors have substantially expanded their supplementary analyses to demonstrate the robustness of their findings in response to the reviewer comments. However, with the text now mentioning all these additional tables, it is harder for readers to follow. For example, the paragraph on page 7 starting with “We can validate...” describes three appendix tables in quick succession and readers may not grasp on first read how these support the CEMP measure. I would recommend revising/reorganizing the description of the appendix tables to not disrupt the flow of the paper so much.

Figure 1 with rates instead of absolute counts is much improved.

Minor:

Typo: The text (page 6) still describes excluding immune-compromised individuals.

Page 16: What is “2H 2021” for Omicron Period?

Table 3 needs to have informative category labels for the age group rather than bins. I see the bins are defined in the caption, but this seems unnecessary when they could just be in the table.

Reviewer #2: All comments have been addressed. I have no further comments other than a few more typos that I have found.

Page 18, line 4. Should be Appendix Table (is currently Rable), and there are two periods at the end of this sentence.

7. PLOS authors have the option to publish the peer review history of their article (what does this mean?). If published, this will include your full peer review and any attached files.

Reviewer #1: No

Reviewer #2: No

---

## [Author Response · Author response to Decision Letter 1]

8 Nov 2023

Please see a separate letter with our replies.

---

## [Decision Letter · Decision Letter 2]

4 Dec 2023

Evidence on COVID-19 Mortality and Disparities Using a Novel Measure, COVID Excess Mortality Percentage: Evidence from Indiana, Wisconsin, and Illinois

PONE-D-23-10973R2

Dear Dr. Yuan,

We’re pleased to inform you that your manuscript has been judged scientifically suitable for publication and will be formally accepted for publication once it meets all outstanding technical requirements.

Kind regards,

Chenfeng Xiong

Academic Editor

PLOS ONE

Additional Editor Comments (optional):

Reviewers' comments:

Reviewer's Responses to Questions

**Comments to the Author**

1. If the authors have adequately addressed your comments raised in a previous round of review and you feel that this manuscript is now acceptable for publication, you may indicate that here to bypass the “Comments to the Author” section, enter your conflict of interest statement in the “Confidential to Editor” section, and submit your "Accept" recommendation.

Reviewer #1: All comments have been addressed

2. Is the manuscript technically sound, and do the data support the conclusions?

Reviewer #1: Yes

3. Has the statistical analysis been performed appropriately and rigorously? 

Reviewer #1: Yes

4. Have the authors made all data underlying the findings in their manuscript fully available?

Reviewer #1: Yes

5. Is the manuscript presented in an intelligible fashion and written in standard English?

Reviewer #1: Yes

6. Review Comments to the Author

Reviewer #1: The authors have addressed all my previous comments. The interpretation of the CEMP in the results section is now clearer for readers. And I appreciate that the authors have discussed the possibility of non-Covid natural mortality increasing during the pandemic for a variety of reasons.

7. PLOS authors have the option to publish the peer review history of their article (what does this mean?). If published, this will include your full peer review and any attached files.

Reviewer #1: No

---

## [Editor Report · Acceptance letter]

18 Jan 2024

PONE-D-23-10973R2 

PLOS ONE

Dear Dr. Yuan, 

I'm pleased to inform you that your manuscript has been deemed suitable for publication in PLOS ONE. Congratulations! Your manuscript is now being handed over to our production team.

Kind regards, 

on behalf of

Dr. Chenfeng Xiong 

Academic Editor

PLOS ONE